# The Influence of Grazing Systems on Bird Species Richness and Density in the Nebraska Sandhills

**Silka L. F. Kempema** [1,*,†], **Walter H. Schacht** [2] **and Larkin A. Powell** [1,*]

1   School of Natural Resources, University of Nebraska-Lincoln, 903 Hardin Hall, 3310 Holdrege Street,
    Lincoln, NE 68583, USA

2   Department of Agronomy and Horticulture, University of Nebraska-Lincoln, 347 Keim Hall,
    Lincoln, NE 68583, USA; wschacht@unl.edu

*   Correspondence: silkalori@hotmail.com (S.L.F.K.); lpowell3@unl.edu (L.A.P.)

†   Current address: U.S. Fish and Wildlife Service, 420 South Garfield Avenue, Suite 400, Pierre, SD 57501, USA.

**Abstract:** Grazing is the de facto method of habitat management used in much of the Nebraska Sandhills. Ranchers use a variety of grazing systems, and our goal was to evaluate the effects of systems on grassland birds. We estimated the species richness and density of grassland birds for three grazing systems used on private ranches: long, medium, and short duration grazing systems. We observed sixty species, and the grazing system with pastures utilizing long duration grazing periods had the highest estimates of species richness as well as the most heterogeneous habitat structure. Differences in species richness among systems were most pronounced in years of limited precipitation. Together, grasshopper sparrows (*Ammodramus savannarum*), western meadowlarks (*Sturnella neglecta*), and brown-headed cowbirds (*Molothrus ater*) accounted for 72% of our observations. We used a model comparison approach to determine the effects of habitat on the densities of six species. Densities of grasshopper sparrows and mourning doves showed effects of the grazing system. More species had higher densities in short duration, rotational systems than other grazing systems. However, species of grassland birds showed responses to a variety of cover types and habitat structures depending on life history needs. Regardless of the grazing system used, managers can use grazing and other tools such as prescribed burning to maintain habitat heterogeneity to support diverse bird communities.

**Keywords:** avian community; density; grasslands; grassland bird; grazing management; great plains; rangelands; species richness





## 1. Introduction

Conservationists in North America have documented negative population trends for breeding grassland birds for decades [1–5]. Annual population declines are steep, consistent, and alarming [6]. Although only 5% of the avifauna in North America is considered grassland endemic [7], these and other grassland-obligate species represent an important component of the continent's biodiversity.

Habitat loss is one reason for grassland bird declines [8]. The temperate grassland biome of North America has experienced precipitous losses through transition to row crop agriculture, and these grassland systems are some of the least conserved [9–11]. For example, the area of habitat lost is as high as 99% for the tallgrass prairie ecosystem [12], and such losses to critical habitats continue [13].

The Sandhills is a 50,000 km$^2$ region of specialized mixed-grass prairie characterized by undulating hills of sandy soils, interspersed with wetlands and subirrigated meadows. The erodible unformed soils have thwarted most efforts to cultivate crops has left much of the region unconverted [14]. As such, it is the largest intact, mixed-grass prairie in North America. The majority of the Sandhills region is privately owned and managed by cattle grazing. Producers use a variety of grazing systems including rotational grazing to give them

flexibility in managing the timing, intensity, and duration of grazing [15]. Rotational grazing may also optimize economic gain by increasing both forage and animal production [16].

Grasslands have been historically maintained by climate, fire and grazing [17,18], and cattle grazing in prairie ecosystems can serve as a surrogate for important yet altered or absent disturbance regimes. However, there is a gap in our understanding of optimal grazing management to benefit species of grassland birds. A complex set of factors including soil conditions, precipitation, habitat type, and grazing intensity affect how livestock grazing influences wildlife response [19,20]. Informed management of privately owned rangeland is critical for grassland bird conservation and maintaining global biodiversity [3,20], yet evidence in the Sandhills region suggests that variation in commonly used grazing systems may not influence the response of bird species and the avian community structure [21].

Given the importance of the Sandhills as habitat for grassland birds and the use of a variety of grazing systems on private lands, we aimed to assess the variation in species richness and density of grassland birds on privately owned rangelands in the Sandhills. The objectives of our study were to (1) estimate species richness and density on pastures under three commonly used grazing systems and (2) evaluate the sources of variation in grassland bird density.

## 2. Materials and Methods

### 2.1. Study Area

We conducted our study on privately owned rangeland in eastern Cherry and northern Thomas counties of the north–central portion of the Nebraska Sandhills, USA, from 2002 to 2004 (Figure 1). The average annual precipitation in this semi-arid region ranges from 432 to 482 mm [22]. Upland soils are Valentine fine sands (mixed, mesic Typic Ustipsamments [23]).

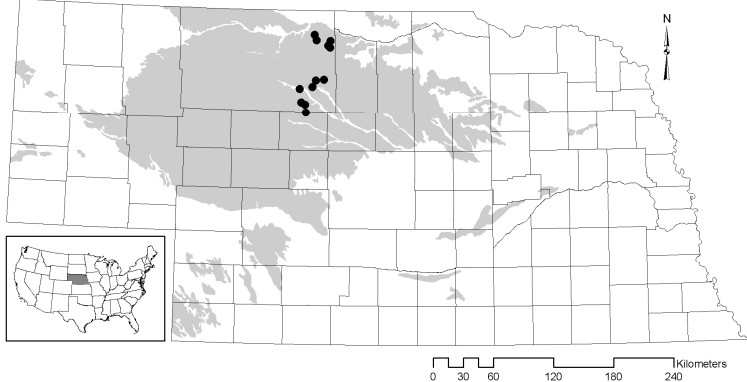

**Figure 1.** Location of study sites (dots) in eastern Cherry and northern Thomas counties in Nebraska, USA. The Sandhills ecoregion is shaded.

We selected participating landowners based upon willingness to participate, range management practices, and management history [24]. We chose only upland sites to reduce any bias in landscape position, topographical differences, or soils associated with wetland and subirrigated meadow sites. We attempted to distribute treatments evenly across the study area.

We evaluated three grazing treatments defined by the length of time a pasture in the system was grazed during the 5-month grazing season (Table 1, Figure 2). Long duration pastures were grazed for an average of 78 days during the grazing season. Medium duration pastures were part of deferred rotation systems with each pasture grazed for an average of 23 days. Short duration pastures were part of grazing systems with 18 pastures or more and with each pasture grazed for an average of 3 days. We documented the stocking rate, defining the animal unit forage demand per unit area (ha) per unit time (month) as animal unit month (AUM/ha), and stocking density, defining the animal unit demand per unit area (ha) at a point in time. Stocking rates were comparable across treatments (Table 1), and

our sample of ranches was selected such that we believe our stocking rates were similar to those commonly used on Sandhills grazing lands. Stocking density is an index of grazing distribution and vegetation heterogeneity. In our study, stocking density increased as grazing duration decreased (Table 1).

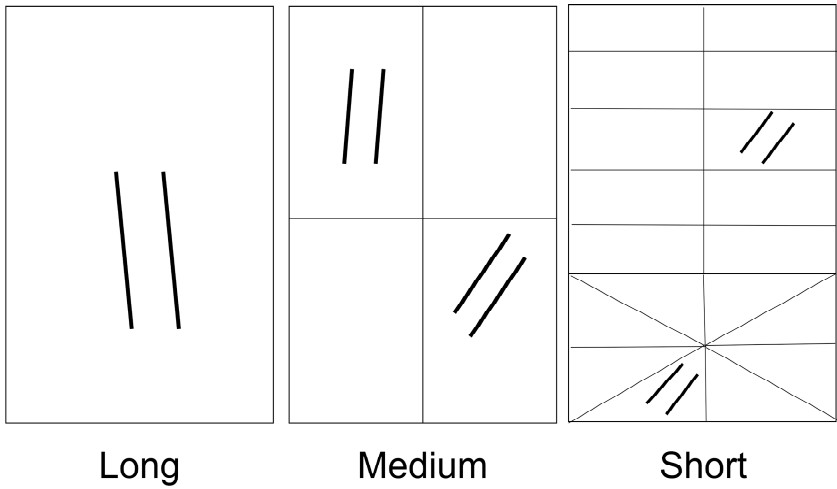

Long                    Medium                    Short

**Figure 2.** Generalized, representative layouts of pastures in the three grazing systems used in this study in eastern Cherry and northern Thomas counties in Nebraska, USA: long duration, medium duration, and short duration. Fences within grazing units created pastures used for rotation of cattle, and grazing systems varied within each category. Parallel lines represent bird survey transects and show examples of how sampling transects were placed within each grazing system (total transect length at each study site in all systems = 3000 m).

**Table 1.** Description of grazing system treatments (long, medium, and short duration) during the growing season (1 May to 30 September) for cattle on private rangelands in the Nebraska Sandhills, 2002–2004. Means and SD are from three replications of each treatment in 2002 and four replicates in 2003–2004. Grazing duration is mean number of days cattle spend in a single pasture. Stocking rate is defined as animal unit months (AUM) per ha, and stocking density is number of animal units per ha at a point in time.

| Descriptor | Long | | Medium | | Short | |
|---|---|---|---|---|---|---|
| | **Mean** | **SD** | **Mean** | **SD** | **Mean** | **SD** |
| Duration (days) | 78 | 26 | 23 | 23 | 3 | 2 |
| Stocking rate (AUM/ha) | 1.4 | 0.96 | 1.32 | 0.68 | 1.42 | 1.27 |
| Stocking density | 0.6 | 0.26 | 2.5 | 2.1 | 11 | 6.3 |
| Herd size | 158 | 47 | 196 | 101 | 626 | 295 |
| Pasture size (ha) | 574 | 239 | 196 | 134 | 82 | 33 |

In 2002, we examined three replications each of long, medium, and short duration grazing systems (Figure 2, Table 1). We added an additional replication per grazing treatment in 2003; thus, a total of 12 study pastures or experimental units were used for the remainder of the study.

## 2.2. Sampling Methods

### 2.2.1. Bird Surveys

We used 3000 m of transect per pasture as our sampling unit. Our sampling intensity and study design were based on an a priori power analysis [25], in which our goal was to minimize the coefficient of variance (CV) for our density estimates. We randomly placed one pair of 1500 m parallel transect lines in each of the long duration pastures, and we placed two pairs of 750 m parallel transects in medium and short duration pastures

(Figure 2). We separated parallel transects by 125 m during 2002 in medium and short duration pastures; transects in long duration pastures were placed 250 m apart. During 2003 and 2004, all transects were separated by 250 m for consistency. We placed each transect pair within separate pastures of medium and short duration systems to ensure we captured effects of the rotational grazing in a staggered manner as cattle moved through the system.

Birds were surveyed during three time periods, or rounds, throughout the summer; the rounds occurred from (1) late May to mid-June, (2) late June to early July, and (3) mid- to late July [24]. During each round, we surveyed all pastures three times on consecutive days, which resulted in nine surveys per pasture per year. We surveyed all treatments during each round, and on a given day, one replication of each grazing system was surveyed. We varied the order in which pastures were surveyed within the three rounds. Before surveys were conducted, observers were trained in sampling protocol [26], distance estimation, and bird identification.

We conducted bird surveys during a 4 to 5 h period beginning at sunrise, when conditions were conducive (<25 km/h winds and limited precipitation). Within a survey round, observers were rotated so that no observer surveyed a pasture twice during that round. Observers used laser rangefinders to record sighting distances and used compasses to determine bearing (nearest degree) to all individuals or clusters of birds seen and/or heard to the nearest meter and degree, respectively [25,27]. We recorded measurements to the initial detection location of the individual or center of the cluster and recorded cluster size. For detections made only by sound, we measured distance to the estimated location of the singing bird. We recorded the method of detection (sight and/or sound) for each bird recorded.

### 2.2.2. Vegetation Surveys

#### 2.2.2.1. Vegetation

We measured the vegetation structure and composition during three rounds of surveys: 11–19 June, 25 June–4 July, and 10–18 July. Vegetation structure was measured at 30 points spaced 100 m apart along survey transects in each pasture. At each point, visual obstruction [28] was read (VOR) to the highest decimeter to which the view of the pole was completely obstructed by vegetation. The percentage cover of vegetation was estimated into eight cover classes: total, dead, live, grass, forb, woody, bare, and litter using a 20 × 50 cm Daubenmire frame [29]. Cactus and yucca (*Yucca glauca*) were included in the woody category. The percentage cover of vegetation categories was estimated into six cover classes: 1 = 0–5%, 2 = 6–25%, 3 = 26–50%, 4 = 51–75%, 5 = 76–95%, and 6 = 96–100%. The depth of litter, defined as vegetation forming a horizontal matte covering the ground, was measured to the nearest cm in the center of the sampling frame at each point. The tallest plant found within the 4 m distance between the visual obstruction pole and the observer was measured and recorded during the second and third rounds of vegetation sampling.

#### 2003–2004 Vegetation

At each 100 m transect interval, including zero, a sampling location was randomly selected within a 15 m radius of the interval for a total of 32 locations in each pasture. Each random location served as the center of an equilateral triangle with 3.5 m sides (sampling unit). The three nodes of the triangle served as sampling points, resulting in 96 total samples for each pasture).

We used a 20 × 50 cm Daubenmire frame [29] to estimate the percentage ground cover at each sampling point. Cover classes included estimates of bunch, rhizomatous, and annual grasses, sedges, forbs, shrubs, cactus, yucca, litter (residual vegetation standing ≤45° to the ground), standing dead (residual vegetation standing at >45° to the ground), bare soil, and cattle dung. We recorded all percentage cover estimates to the nearest 5%, allowing for total cover classes to exceed 100%. The estimated cover of bunch and rhizomatous grasses were pooled to create a perennial grass cover category. We recorded

VOR to the nearest quarter decimeter from four directions at right angles [28] from a square pole. An average of these four readings was used to describe the vegetation height and density at each sampling point.

We incorporated our VOR means in a modified sampling protocol [30] that allowed us to quantify the structural heterogeneity of the grassland bird habitat at broad (pasture-level) and fine (patch-level) spatial scales. We refer to these scales as large- and small-scale heterogeneity. The mean small-scale heterogeneity (*SSH*) for each pasture, *r*, was calculated as:

$$SSH_r = \frac{\left( \sum_{i=1}^{32} \left( Max_{k_i} - Min_{k_i} \right) \right)}{\sum_{i=1}^{32} \overline{x}_{k_i}}$$

where *i* = 1, 2, . . ., 32 for each sampling location, while *k* = 1, 2, 3 (three sub-samples taken at *i*), which represents the mean of VOR scores from three sub-samples at *i*. We used the CV to represent large-scale heterogeneity (*LSH*) for each pasture, *r*:

$$LSH_r = \frac{\sigma}{\overline{x}_i} \times 100$$

where $\sigma = SD\left( \overline{x}_{k_i} \right)$.

We measured and recorded the height of the tallest plant (to the nearest 0.25 dm) located within a 30 cm radius of the sample point, its functional group (grass, forb, shrub, sedge, rush or yucca), and status (alive or dead). Litter depth was measured to the nearest 0.5 cm within 3 cm of the upper right-hand corner of the sampling frame. We also estimated the percentage cover of all plants based upon origin (native or exotic).

### 2.3. Statistical Analysis

#### 2.3.1. Species Richness Estimates

We obtained estimates of bird species richness using the program SPECRICH2 [31,32]. Program SPECRICH2 uses a closed mark–recapture model to address incomplete species detectability. Species richness estimates are determined based upon presence–absence data from multiple sites or visits: in this instance, multiple pastures (i = 1, 2 . . .k; k = 3 in 2002 and k = 4 in 2003 and 2004) of each grazing system. We recorded (1) the total number of species observed at each pasture for all pastures in each of three grazing systems, $n_i$, and (2) the total number of species observed at k pastures is recorded. A species is 'captured' and 'marked' when it is initially observed during a visit, and additional detections of the same species are treated as 'recapture' events. We calculated the 3-year means for species richness estimates for each grazing system, and we used the delta method [33] to calculate the confidence intervals for the mean estimates.

#### 2.3.2. Density Estimates

We calculated the relative abundance (total number of individuals observed/total survey effort, (birds/m)) for all species observed. We used the program Distance (v5.0, Beta 5) to estimate densities for each species with a sample size of ≥60 individuals per year [25]. Density estimates were calculated using line transect data analyzed as clusters; cluster size estimation compensated for size bias by regressing the natural log (ln) of cluster size against the estimated detection function, g(x). We pooled all nine surveys within the same year. Our density estimates, therefore, are representative of the grazing system during the breeding season.

We visually inspected the frequency distribution of the observed perpendicular distances to determine the utility of binning the data [25]. Frequency histograms of the perpendicular distance data by species within year did not suggest data heaping (i.e., distance measurement rounding errors); therefore, data were not binned. We also used visual

inspection to remove outliers through the right truncation of the data. We applied species- and year-specific truncation distances [24].

We tested for an observer bias each year for species that had $\geq 60$ observations by post-stratifying the data based upon observer and selecting among a five-model set using information theoretic selection criteria.

We initially estimated density for each species at three hierarchical levels: sample (12 pastures), strata (three grazing systems), and global (all samples) [25]. We calculated global density estimates as an average of the stratum estimates weighted by total effort in strata, and we treated strata as a fixed effect.

A model selection process was conducted within each of the three levels to select the most appropriate shape for the detection function from among five standard models used in distance sampling. The three key functions, or standard detection curve shapes, were half normal, uniform, and hazard rate, and the three adjustment terms to the detection functions were cosine, hermite polynomial, and simple polynomial [25]. We used five competing models for the shape of detection function: half normal key function with cosine and hermite polynomial adjustment terms, uniform key function with cosine and simple polynomial adjustment terms and the hazard rate key function with the cosine adjustment term. Program Distance evaluated the models using Akaike's Information Criterion (AIC). We selected the model with the lowest AIC score as the best model to explain the shape of species-specific detection functions. We evaluated the model fit using the Kolmogorov–Smirnov goodness-of-fit test ($\alpha = 0.05$) [25].

The detection of birds can potentially vary among samples or strata because of habitat features. We used model comparison (AIC) to determine if detection varied among the geographic strata. Once we selected the most appropriate basic detection model (shape and geographic strata), we evaluated two additional factors that had the potential to affect the detectability of grassland birds. We modified the best basic, species-specific model, and compared the modified models to the basic model using AIC. First, we created a model that estimated two detection functions: auditory (observed with sound only) and visual (seen or seen/heard) detection of birds. A second model accounted for potential differences in detection for birds initially observed on fences or fence posts. We collected this covariate information only during 2003–2004. We used a similar model selection process (AIC) within each of the covariate models. We combined two key functions and three adjustment terms to create a five-model set, consisting of the half normal key function with three adjustment terms (cosine, simple, and hermite polynomial) and the hazard rate key function with two adjustment terms (cosine and simple polynomial). We used the multiple covariates sampling engine in the program Distance for this process [25].

We used density estimates from the best model for each species [24]. We used bootstrapping (999 resamples) to determine measures of variance for those species in which the sample or strata level detection functions were used for density estimation [25]. We used 95% confidence intervals to compare species density among systems within years and among years within a particular system.

To explain the variability in density among study sites, we created an a priori, limited set of single-factor models based on hypothesized causal mechanisms from literature sources and our observations. We limited this analysis to the most abundant species for which we could estimate density for three years, and we used a separate model selection process for each species. Competing models included the grazing system (SYST), vegetation structural components, and growing season stocking rate (GsSR) and stocking density (SDEN). We selected the best models using AICc values (AIC, modified for small samples) [34] generated in PROC MIXED [35]. We treated year as a random effect.

## 3. Results

### 3.1. Species Richness

We recorded 60 species of birds during our study (2002: 32, 2003: 53, 2004: 49); 28 species were observed in all three years (Table 2). We observed 35, 34, and 28 species on

long, medium, and short duration systems, respectively. Bird species richness estimates tended to be higher on pastures with long duration grazing systems and shorter on pastures with short duration grazing systems (Figure 3). Mean species richness estimates were 52.7 (SE = 4.0), 47.7 (SE = 3.7) and 36.3 (SE = 2.8) species for long, medium, and short duration grazing systems, respectively.

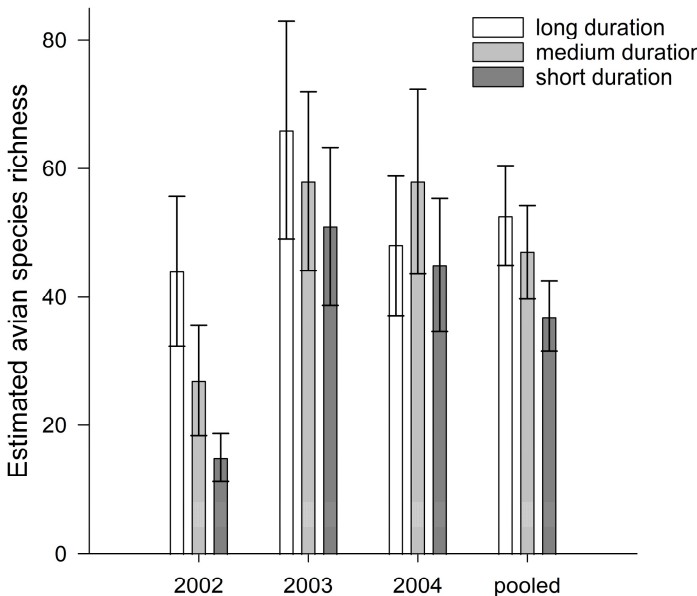

**Figure 3.** Estimates of bird community species richness, accounting for incomplete detectability, in three grazing systems (long, medium, and short duration) on private lands in the Nebraska Sandhills, 2002–2004 and the means pooled across years. Error bars show 95% confidence intervals.

**Table 2.** Twenty-eight species of birds observed during three years of study on private lands in the Nebraska Sandhills, 2002–2004. Six species used in density analyses are shown with *.

| Common Name | Scientific Name |
|---|---|
| American crow | *Corvus brachyrhynchos* |
| American goldfinch | *Carduelis tristis* |
| Bell's vireo | *Vireo bellii* |
| Brown-headed cowbird * | *Molothrus ater* |
| Brown thrasher | *Toxostoma rufum* |
| Common nighthawk | *Chordeiles minor* |
| Dickcissel | *Spiza americana* |
| Eastern kingbird | *Tyrannus tyrannus* |
| Field sparrow | *Spizella pusilla* |
| Greater prairie-chicken | *Tympanuchus cupido* |
| Grasshopper sparrow * | *Ammodramus savannarum* |
| Horned lark | *Eremophila alpestris* |
| Killdeer | *Charadrius vociferus* |
| Lark bunting | *Calamospiza melanocorys* |
| Lark sparrow * | *Chondestes grammacus* |
| Long-billed curlew | *Numenius americanus* |
| Mallard | *Anas platyrhynchos* |
| Mourning dove * | *Zenaida macroura* |
| Northern harrier | *Circus hudsonius* |
| Orchard oriole | *Icterus spurius* |
| Ring-necked pheasant | *Phasianus colchicus* |
| Red-winged blackbird | *Agelaius phoeniceus* |
| Spotted towhee | *Pipilo maculatus* |

**Table 2.** *Cont.*

| Common Name | Scientific Name |
|---|---|
| Sharp-tailed grouse | *Tympanuchus phasianellus* |
| Upland sandpiper * | *Bartramia longicauda* |
| Western kingbird | *Tyrannus verticalis* |
| Western meadowlark * | *Sturnella neglecta* |
| Willet | *Catoptrophorus semipalmatus* |

### 3.2. Abundance and Density

We recorded 31,685 observations of birds by sampling 890,100 m of transect [24]. Three species accounted for ~72% of our observations: grasshopper sparrow (*Ammodramus savannarum*, 31.6%), western meadowlark (*Sturnella neglecta*, 30.3%), and brown-headed cowbird (*Molothrus ater*, 10.4%). Other species composing at least 1% of the total sample included lark sparrow (*Chondestes grammacus*, 7.6%), upland sandpiper (*Bartramia longicauda*, 3.5%), horned lark (*Eremophila alpestris*, 3.1%), mourning dove (*Zenaida macroura*, 2.4%), vesper sparrow (*Pooecetes gramineus*, 1.5%), lark bunting (*Calamospiza melanocorys*, 1.4%), and red-winged blackbird (*Agelaius phoeniceus*, 1.0%).

There was not a consistent observer effect on detection [24], so we combined data from observers for density estimation. We estimated the densities of six species for which a sufficient sample size ($n \geq 60$) was available in all study years: brown-headed cowbird, grasshopper sparrow, lark sparrow, mourning dove, upland sandpiper, and western meadowlark (Table 3).

**Table 3.** Density estimates (birds/100 ha) for six species on three grazing systems (long, medium, and short duration) on private rangeland in the Nebraska Sandhills, 2002–2004. Pooled density estimate is across grazing systems, representing species density in our study area in the Sandhills region. *p*-value is from testing for the effect of the grazing system on species' density estimates from the grazing systems model in the mixed model analysis.

| Species | Grazing System | | | | *p*-Value |
|---|---|---|---|---|---|
| | Long | Medium | Short | Pooled | |
| Brown-headed cowbird | 22.2 | 6.2 | 33.4 | 20.6 | 0.111 |
| Grasshopper sparrow | 57.0 | 114.6 | 97.3 | 89.7 | 0.047 |
| Lark sparrow | 24.2 | 12.6 | 17.6 | 18.1 | 0.151 |
| Mourning dove | 6.4 | 3.7 | 6.8 | 5.6 | 0.010 |
| Upland sandpiper | 5.7 | 5.2 | 5.5 | 5.5 | 0.899 |
| Western meadowlark | 29.1 | 39.3 | 42.0 | 36.8 | 0.065 |

Study ranches supported a mean of 20.6 brown-headed cowbirds/100 ha, making its density the third highest among all species. Densities of this species were highly variable; grazing duration did not contribute to variability. Variation in brown-headed cowbird densities was explained best by vegetation height (AICc = 235.3, $w_i$ = 0.611) and bunchgrass cover (AICc = 0.9, $w_i$ = 0.389). Although ranked above the null model, neither effect had model coefficients significantly different from zero (Table 4).

Grasshopper sparrow density was the highest of any species in our study (89.7 birds/100 ha) and varied among grazing systems (Table 3, $p$ = 0.047) with lowest densities on long duration systems (Table 3). Shrub cover (AICc = 246.1, $w_i$ = 0.65) and early large-scale heterogeneity (ΔAICc = 1.2, $w_i$ = 0.35) explained the most variation in grasshopper sparrow density. Grasshopper sparrow densities increased with shrub cover ($\beta$ = 4.11, SE = 6.47) and declined as large-scale heterogeneity increased ($\beta$ = −0.75, SE = 0.31). Lower-ranked models provided some evidence that the grasshopper sparrow density was higher at plots with thicker cover (VOR $\beta$ = 196.9, SE = 51.58) and deeper litter cover ($\beta$ = 266.65, SE = 61.12; Table 4).

**Table 4.** Results of competing model comparisons for a priori, species-specific models proposed to describe variation in grassland bird density observed on private rangeland in the Nebraska Sandhills, 2002–2004. Except for null, each model has two parameters (intercept and main effect, *β*). Models with lower ΔAIC values and higher Akaike weights have more support given the data and the model set. Models with $w_i$ > 5% are shown, as well as the null model (no effect). The grazing system models have system-specific values for *β* not shown here.

| Species | Model [a] | AICc | ΔAICc | $w_i$ | $β$ | SE |
|---|---|---|---|---|---|---|
| Brown-headed cowbird | Tallest plant | 235.3 | 0.0 | 0.611 | −1.15 | 1.48 |
| | Bunchgrass cover | 236.2 | 0.9 | 0.389 | −0.04 | 1.30 |
| | Null | 333.4 | 98.1 | 0.000 | - | - |
| Grasshopper Sparrow | Shrub cover | 246.1 | 0.0 | 0.646 | 4.11 | 6.47 |
| | Large-scale heterogeneity (early season) | 247.3 | 1.2 | 0.354 | −0.75 | 0.31 |
| | Null | 354.6 | 108.5 | 0.000 | - | - |
| Lark sparrow | Shrub cover | 192.5 | 0.0 | 0.917 | 1.63 | 1.91 |
| | Litter cover | 197.3 | 4.8 | 0.083 | −0.57 | 0.24 |
| | Null | 268.6 | 76.1 | 0.000 | - | - |
| Mourning dove | Grazing system | 154.7 | 0.0 | 0.996 | - | - |
| | Null | 167.9 | 13.2 | 0.001 | - | - |
| Upland sandpiper | Shrub cover | 133.3 | 0.0 | 0.973 | −1.06 | 0.47 |
| | Null | 186.2 | 52.9 | 0.000 | - | - |
| Western meadowlark | Bunchgrass cover | 195.2 | 0.0 | 0.731 | 1.52 | 0.59 |
| | Shrub cover | 198.4 | 3.2 | 0.148 | 0.26 | 1.99 |
| | Large-scale heterogeneity (late season) | 198.8 | 3.6 | 0.121 | −0.27 | 0.11 |
| | Null | 274.3 | 79.1 | 0.000 | - | - |

[a] Models with $w_i$ < 5% not shown in detail are (1) brown-headed cowbird: litter depth, VOR, system, growing season stocking rate, forb cover, stocking density; (2) grasshopper sparrow: VOR, grazing system, litter depth, growing season stocking rate; (3) lark sparrow; grazing system, growing season stocking rate, bare soil cover; (4) mourning dove: growing season stocking rate; (5) upland sandpiper: large-scale heterogeneity (early season), grazing system, VOR, growing season stocking rate, bare soil cover; (6) western meadowlark: VOR, litter depth, grazing system, growing season stocking rate, forb cover.

Lark sparrow density was 18.1 birds/100 ha (Table 3). The shrub model was the highest ranked, although we did not find strong evidence to suggest that thicker shrub cover caused higher densities of lark sparrows (AICc = 192.5, $w_i$ = 0.917; $β$ = 1.63, SE = 1.91). Lower-ranked models provided some evidence that lark sparrow densities decreased with more litter cover ($β$ = −0.566, SE = 0.24), increased with more bare soil ($β$ = 0.537, SE = 0.17), and decreased with growing season stocking rate ($β$ = −5.728, SE = −2.42; Table 4).

Mourning dove density was 5.6 birds/100 ha. The grazing system model was ranked the highest and provided evidence that the highest densities were found on short and long duration grazing systems (AICc = 154.7, $w_i$ = 0.996; $p$ = 0.010, Tables 3 and 4).

Upland sandpiper density was 5.47 birds/100 ha on our study sites, and we observed no difference in density among grazing systems (Table 3). Shrub cover explained the most variation in upland sandpiper density (AICc = 133.3, $w_i$ = 0.917; Table 4), which decreased as shrub cover increased ($β$ = −1.06; SE = 0.47).

Western meadowlark density on the study ranches averaged 36.8 birds/100 ha (Table 3). Although Western meadowlark densities varied among grazing systems, this model was not ranked among the top models. The bunchgrass model (AICc = 195.2; $w_i$ = 0.731) explained the most variation in western meadowlark densities, which increased as bunchgrass cover

increased ($\beta$ = 1.52, SE = 0.59; Table 4). Lower-ranked models provided some evidence that meadowlark densities increased with thicker cover (VOR $\beta$ = 54.52, SE = 15.15) and deeper litter depth ($\beta$ = 54.86, SE = 17.20). Densities decreased on sites with higher late-season, large-scale heterogeneity ($\beta$ = −0.27, SE = 0.11).

## 4. Discussion

### 4.1. Grazing Systems

It is challenging to draw simple conclusions regarding the effects of grazing systems on the grassland bird community in the Nebraska Sandhills. The mean bird species richness estimates across years in our study were highest in long duration pastures, and richness decreased with shorter grazing duration. However, our observations in 2004 did not follow this trend (Figure 3).

Densities of grasshopper sparrows, mourning doves, and western meadowlarks varied with grazing system, but no consistent response was observed among these species. Our results indicate that the grazing duration associated with commonly used grazing systems in the Nebraska Sandhills influences grassland bird species richness and the density of some species. However, the grazing intensity among systems was similar in our study, and our analysis lends some support to previous work [36,37] that suggested grazing systems may have relatively low importance to songbird communities in the Great Plains. Instead, our results suggest that local habitat structure, however derived, may be more important.

Although we classified our treatments as long, medium and short duration grazing systems, there was variation in the grazing duration, stocking rate, and stocking density within each of the treatments [21,37]. This reflects that our research was conducted on privately owned and operated ranches where each manager determines grazing parameters that best fit their situation and goals. It is no surprise, therefore, that other on-ranch research shows a low level of agreement on grazing system effects. For example, grasshopper sparrow densities were similar among grazing system treatments in a North Dakota study [38], higher on rotational systems in a second North Dakota study [39], and highest on long duration (season-long continuous) systems in Nebraska [40].

Grazing [41–43] and the grazing system may elicit different responses from different grassland birds. These responses also may vary depending on grassland types [44], making the interpretation of the relationships between grazing and wildlife complex [45].

The mean percentage vegetation cover did not differ among the grazing system pastures in our study [24]. However, grazing systems on our study sites exhibited differences in structural heterogeneity (variability in biomass as measured by VOR). The vegetation structures of medium and short duration grazing systems were relatively more homogeneous compared to long duration systems at both a small and large spatial scale [24]. But ranchers in our study used their grazing system in a generally conservative manner to provide for sustainable forage levels for their cattle, in effect using different tools to achieve the same goal [46]. It is not surprising, therefore, that we found no robust, consistent effects of our grazing systems on densities of grassland birds in the Sandhills region.

### 4.2. Critical Vegetation

Vegetation structure, including shrub cover, grass cover, VOR, and litter depth, was useful to explain variation in avian species densities among study sites. And typically, vegetation-based models were ranked higher than the grazing systems model for our six species of birds. Grassland birds are known to respond to habitat structure [47–49], and vegetation structure varied considerably within grazing system in our study [24].

Shrub cover on our pastures was low (5%) [24]; however, three key species responded to variations in shrub cover. Grasshopper sparrow and lark sparrow densities increased with shrub cover, and upland sandpiper densities were inversely related to shrub cover. Suitable habitat for upland sandpipers [50] and grasshopper sparrows [51] usually contains low amounts of woody cover. Lark sparrows use the base of shrubs as a nesting substrate [52,53].

Grasshopper sparrow densities also were positively associated with VOR and litter depth, but they were negatively related to early large-scale heterogeneity. In North Dakota, grasshopper sparrow densities were positively associated with these same measurements [38]. Grasshopper sparrow abundance has been found to be positively correlated with litter depth [48]; furthermore, the same study reported that grasshopper sparrows decreased as forb and shrub heights became more variable.

Western meadowlark density increased with bunchgrass cover, VOR and litter depth and decreased with large-scale heterogeneity during August. This response is similar to grasshopper sparrows by having a positive relationship with VOR and litter depth. Previous work reported a positive relationship between meadowlark densities and grass cover and litter depth [38]. Suitable habitat for western meadowlarks encompasses a variety of vegetation heights and densities with the avoidance of extremely sparse or tall vegetation [54].

Brown-headed cowbird density was positively related to plant height and negatively related to bunchgrass cover, which may reflect both their breeding and foraging habits, respectively. Taller vegetation provides perch sites to search for host nests, and reduced grass cover facilitates ground foraging behavior. Previous research also showed a negative relationship between cowbird densities and grass cover [55]. However, research examining the effects of rotational and traditional season long grazing systems on grassland birds reported that brown-headed cowbird densities were positively related to grass cover, litter depth and shrub cover [38].

In addition to the shrub cover mentioned above, lark sparrow density was positively related to bare soil and litter. The density of this sparrow was negatively related to growing season stocking rate. Bare ground is important for foraging habitat, and biologists recommended moderate grazing for lark sparrows [53]. Although lark sparrows tend to select grazed areas [56], our data suggest that high levels of grazing could reduce lark sparrow densities. Management for this species should include moderate levels of litter accumulation and bare ground [56].

The Nebraska Sandhills is a unique landscape with more contiguous grasslands than in other regions of North America [57]. The size of the Sandhills region creates the potential for a high diversity of habitat structures throughout the region, which would support a variety of grassland birds. We did not find any of the grazing systems in our study to be detrimental to grassland birds; all systems appeared to support a diverse grassland bird community. Ranchers in our study used appropriate stocking rates for the semi-arid region [24].

## 5. Conclusions

Our data suggest that the conservation and management of grassland birds in the Nebraska Sandhills are not affected by the range of grazing duration of the grazing systems used in this study. Ranchers typically manage for efficient use, temporally and spatially, of the forage resource for cattle despite the grazing system used. Thus, management aimed at the conservation of grassland birds should start with site-specific goals that target a particular suite of specific grassland species. Long duration systems tended to provide greater species richness, while densities of specific species were maximized in different systems. Managers' goals should be framed within a landscape context, and we encourage managers of public and private lands to consider local and landscape-level habitat heterogeneity as part of a healthy, functioning ecosystem [58,59]. Grassland birds evolved in landscapes with large-scale habitat heterogeneity, and our study suggests that such variation in habitat is influenced by topography, responses to drought, and grazing intensities.

Our study also suggests that grassland bird conservation and profitable range management are not incompatible. We found comparable grassland bird populations across landowners and grazing systems. Avian conservation and animal management in semi-arid regions, like the Nebraska Sandhills, will accrue long-term benefits through the prevention of overgrazing and drought management. Grassland birds in our study showed responses

to a variety of cover types and habitat structures, depending on life history needs. Regardless of the grazing system used, the use of grazing and other tools such as prescribed burning can maintain landscape heterogeneity.

**Author Contributions:** Conceptualization, L.A.P., W.H.S.; methodology, L.A.P., S.L.F.K., W.H.S.; formal analysis, S.L.F.K.; resources, L.A.P.; data curation, S.L.F.K.; writing—original draft preparation, S.L.F.K.; writing—review and editing, L.A.P., W.H.S.; visualization, L.A.P., S.L.F.K., supervision, L.A.P., S.L.F.K.; project administration, L.A.P.; funding acquisition, L.A.P. All authors have read and agreed to the published version of the manuscript.

**Funding:** This research was funded by the Nebraska Sandhills Taskforce, the Nebraska Game and Parks Commission, Sampson Range and Pasture Endowment, and a United States Department of Agriculture Sustainable Agriculture and Research Education grant.

**Institutional Review Board Statement:** The animal study protocol was approved by the University of Nebraska-Lincoln's Institutional Animal Care and Use Committee (#04-03-018; approved March 2002).

**Data Availability Statement:** Data presented in this study are available through the University of Nebraska-Lincoln Data Repository: https://doi.org/10.32873/unl.dr.20231018 (accessed on 20 November 2023).

**Acknowledgments:** Several people and organizations made this research possible. Foremost, we are grateful to eight private landowners who allowed us to access their lands for three years. K. Elder, K. Fricke, J. Jordening, S. Larson, J. Milliken, A. (Noe) Reade, A. Richardson, and K. Unstad provided assistance collecting field data; M. Proett provided valuable work during the pilot year of the study. E. Blankenship and R. Johnson provided technical guidance for project design. The School of Natural Resources and the Department of Agronomy and Horticulture provided support for this project. LAP and WHS were supported by Hatch Act funds through the University of Nebraska Agricultural Research Division, Lincoln, Nebraska.

**Conflicts of Interest:** The authors declare no conflict of interest. The funders had no role in the design of the study; in the collection, analyses, or interpretation of data; in the writing of the manuscript; or in the decision to publish the results.

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
