# Peer review of "The Influence of Grazing Systems on Bird Species Richness and Density in the Nebraska Sandhills"

_diversity, doi:10.3390/d15121160_

Round 1

Reviewer 1 Report

Comments and Suggestions for Authors

Review comments/questions:

-          Line 45, tense problem – soils have vs has left, rephrase.

-          Line 61, add the before Sandhills to be consistent with how you refer to it elsewhere (i.e., Sandhills vs the Sandhills).

-          Line 96, line numbers are over top of the table 1 text.

-          Lines 148-49, is 125m apart sufficient to distinguish separate birds on parallel transects in 2002?

-          Lines 166-70, if I am understanding correctly, you record distance to a bird whether seen or heard?  How do you record distance to a bird detected only by sight? If you don’t see the bird, do I assume it’s the cluster you are measuring distance to?

-          Line 220, several paragraphs before this are all about vegetation, can you clarify here if you’re estimating species richness of plants or birds in this next section? I assume birds but it is a bit confusing given the preceding methods.

-          Lines 254-59, this is confusing.  I am familiar with detection relating to time of day, temperature, observer, etc, but I am not familiar with half normal key function, cosine and hermite polynomial, uniform key function, etc.  Can you explain/cite in a sentence or two what this is attempting to do?

-          Line 283, remove “was done” from the end of this sentence?

-          Line 319, remove D in front of upland sandpiper.

-          Line 350, period missing between density and Grasshopper.

-          Line 373, remove with from this sentence.

-          Table 4, what is lying litter cover?  Is there litter that is not lying?

-          Lines 400-405, I am somewhat confused by the logic flow in this paragraph.  Species richness is higher on long duration, densities of 3 species vary with grazing system but not in the same way, and therefore grazing systems are not important?  Do you imply they are not important because species do not all respond in the same way?

-          Lines 406-411, similarly confusing in the way this is all phrased.  When you get to 413-415, it seems this is really the crux of the problem/argument and perhaps you should lead with some version of this statement.  It seems that, although it is challenging to draw conclusions across many species responding differently, that does not necessarily mean that grazing is not important, but instead that it influences species differently from one another.

-          Line 447-448, extremes are avoided by the birds?  Meaning uniform characteristics?  All tall or all short?  Extreme as a contrast to variety is somewhat confusing terminology.

-          Line 451, provides instead of proves I assume you mean.

-          Line 481-484, again the logic escapes me slightly.  Do you mean, in the first part of the sentence, that you would expect ranchland managers to create a variety of structures across the landscape such that they could emulate historical conditions? 

-          Line 486, do you mean diametrically opposed?

-          Line 485-492, I appreciate these concluding thoughts and wonder if you might say anything about the role that grazing plays in influencing nest success for any of these species?  Although it appears that their habitat needs can be met among a variety of rotations as measured by their richness and density, what does the literature suggest about the compatibility of grazing systems with their ability to successfully reproduce in these fields?  Are there any lessons there?  Maybe just a sentence or two to note that component would be a nice addition. 

Author Response

Replies to Reviewer #1

- Line 45, tense problem – soils have vs has left, rephrase.

        --rephrased as suggested

-          Line 61, add the before Sandhills to be consistent with how you refer to it elsewhere (i.e., Sandhills vs the Sandhills).

        --modified throughout as requested.

-          Line 96, line numbers are over top of the table 1 text.

        --understood, we believe this was a formatting issue in the PDF creation, and will make sure during finalization of the manuscript that this feature is not a problem.

-          Lines 148-49, is 125m apart sufficient to distinguish separate birds on parallel transects in 2002?

--we agree and was the reason we changed in 2003. For analysis, it does not impact density estimates because the potential for double sampling the same bird happens independently for distance-type data used to estimate density; however we realized our design lacked efficiency and we could cover more space by increasing the distance between transects to 250 meters. We added some wording to clarify in the text.

-          Lines 166-70, if I am understanding correctly, you record distance to a bird whether seen or heard?  How do you record distance to a bird detected only by sight? If you don’t see the bird, do I assume it’s the cluster you are measuring distance to?

        --Yes, we measure distance to the estimated location, based on training and field experience. We clarified in the text.

-          Line 220, several paragraphs before this are all about vegetation, can you clarify here if you’re estimating species richness of plants or birds in this next section? I assume birds but it is a bit confusing given the preceding methods.

        --Clarified in the text as requested. Yes, it is birds.

-          Lines 254-59, this is confusing.  I am familiar with detection relating to time of day, temperature, observer, etc, but I am not familiar with half normal key function, cosine and hermite polynomial, uniform key function, etc.  Can you explain/cite in a sentence or two what this is attempting to do?

        --Yes, and we added quite a bit of text to clarify for those not familiar with the distance-sampling techniques. Program Distance was developed to estimate density based on distributions of observations away from the transect. As distance from transect increases, detection probability decreases.  However, the rate at which detection probability decreases is different for each species and situation. So, a half normal key function uses the mathematical equation for a normal distribution to fit a half normal decay function to the observed data. Uniform has no decay. Hazard has a precipitous decay function. The adjustment terms allow those basic decay functions to be tweaked (creating a more complicated decay function equation) to try to fit the pattern of the observed data better.

-          Line 283, remove “was done” from the end of this sentence?

        --Removed as requested.

-          Line 319, remove D in front of upland sandpiper.

        --Fixed as requested.

-          Line 350, period missing between density and Grasshopper.

        --Fixed as requested.

-          Line 373, remove with from this sentence.

--Fixed as requested.

-          Table 4, what is lying litter cover?  Is there litter that is not lying?

        --We removed ‘lying’ so that it just reads “litter cover”. This now matches the description in the methods. We sometimes using “lying litter” to refer to dead vegetation with stems <45 degrees from the ground and “standing little” to refer to standing (stems >45 degrees) dead vegetation. But that was not a useful distinction in the manuscript.  

-          Lines 400-405, I am somewhat confused by the logic flow in this paragraph.  Species richness is higher on long duration, densities of 3 species vary with grazing system but not in the same way, and therefore grazing systems are not important?  Do you imply they are not important because species do not all respond in the same way?

        --We have modified this section extensively. We were attempting to suggest that there is not a preferred grazing system that benefits all species in the same manner.

-          Lines 406-411, similarly confusing in the way this is all phrased.  When you get to 413-415, it seems this is really the crux of the problem/argument and perhaps you should lead with some version of this statement.  It seems that, although it is challenging to draw conclusions across many species responding differently, that does not necessarily mean that grazing is not important, but instead that it influences species differently from one another.

        --We have rewritten this section to respond to your suggestion.

-          Line 447-448, extremes are avoided by the birds?  Meaning uniform characteristics?  All tall or all short?  Extreme as a contrast to variety is somewhat confusing terminology.

        --We clarified this sentence.

-          Line 451, provides instead of proves I assume you mean.

        --Spelling fixed as suggested.

-          Line 481-484, again the logic escapes me slightly.  Do you mean, in the first part of the sentence, that you would expect ranchland managers to create a variety of structures across the landscape such that they could emulate historical conditions? 

        --We agree this was not clear. We have rewritten a portion of this paragraph.

-          Line 486, do you mean diametrically opposed?

        --We changed phrasing to read ‘not incompatible’ to be simpler.

-          Line 485-492, I appreciate these concluding thoughts and wonder if you might say anything about the role that grazing plays in influencing nest success for any of these species?  Although it appears that their habitat needs can be met among a variety of rotations as measured by their richness and density, what does the literature suggest about the compatibility of grazing systems with their ability to successfully reproduce in these fields?  Are there any lessons there?  Maybe just a sentence or two to note that component would be a nice addition. 

        --We appreciate the positive comments here, and the interest in nest success. In this paper, we have not focused on productivity/nest survival. We have data on nest survival in a manuscript in preparation from this same study, and we felt that in the conclusion was not the best place to introduce this new topic. But we agree it is an important component of the equation.

Reviewer 2 Report

Comments and Suggestions for Authors

Abstract:

Lines 16-17:  Please provide the three grazing systems that were evaluated:  “We estimated species richness and density of grassland birds for three grazing systems used on private ranches: long, medium, and short duration rotational grazing systems.”

Lines 20-21:  Please provide scientific names for grasshopper sparrow, western meadowlark, and brown-headed cowbird.

Line 21:  The authors provided a value for the percentage of all observations that comprised the three most common species, but there is a discrepancy on the percentage (70%) given on Line 21 in the Abstract and the percentage (approximately 75%) given on Line 316 in the Results.  Based on the additional information provided in the Results, the percentage is closer to 72.3%.  I recommend including this value in both places.

Line 22:  The wording (6 species’ density) at the end of this sentence is somewhat awkward.  I recommend rewording this sentence:  “We used a model comparison approach to determine the effects of habitat on the densities of six species.”

Introduction

Lines 45-46.  The sentence ends somewhat awkwardly. I believe that the second to last word of the sentence should be singular (i.e., region rather than regions):  “The erodible unformed soils have thwarted most efforts to cultivate crops and has left much of the region unconverted.”

Line 57 and 63:  Do not include a hyphen between “privately” and “owned.”  Compounds formed by an adverb ending in “ly” plus an adjective or participle (such as chiefly unrelated or neatly organized) are not hyphenated either before or after a noun.

Methods

Line 69: Do not include a hyphen between “privately” and “owned.”  Compounds formed by an adverb ending in “ly” plus an adjective or participle (such as chiefly unrelated or neatly organized) are not hyphenated either before or after a noun.

Line 128:  For consistency, write “3” as “three.”  For example, see Lines 137 and 139.

Line 139:  For consistency, write “twelve” as “12.” Numbers over 10 are likely written as numerals.

Line 146: For consistency, include a comma for “1,500-m” as done on Line 143.

Line 162:  Change the following “four-to-five hour period” to “45-hour period” because hour is a unit.

Line 179:  Provide the scientific name for “yucca.”

Line 181:  Change “Litter” to “The depth of litter.”

Line 199:  Write out “dm” to “decimeter.”

Line 210:  Delete “coefficient of variation” and just use “CV.”  Coefficient of variation was already defined on Line 45.

Lines 229-230:  Change “three-year means” to “3-year means” because year is a unit.

Line 255: Change “5” to “five” for consistency.

Results

Line 295:  Change “three years” to “3 years” because year is a unit.

Lines 296-297:  The sentence on these lines indicate that “Species richness estimates decreased with grazing duration (Fig. 3).”  In actuality (and as shown in the sentence on Lines 297-299 and Figure 3), species richness increased with grazing duration.  That is, for example, species richness was lower in short duration grazing systems (36.3 species) compared to long duration grazing systems (52.7 species).

Line 316:  See comment above concerning “75%.”

Line 319:  Change “Dupland sandpiper” to “upland sandpiper.”

Line 350:  Insert a period after “density.”

Lines 364:  Should there be the word “and” between “Tables 3,4”?  That is, should it be written as “Tables 3 and 4?”

Line 373:  Delete “with.”

Discussion

Line 400:  Clarify.  The authors wrote: “Species richness tended to be higher on long duration study sites,…” Please clarify higher than what or compared to what.

Line 406:  It is not clear what the authors mean by “meaningful ways among the same category.”  What is a meaningful way? What category are the authors referencing?  The same goes for Line 407.  What is a grazing system category? “Category” or “categories” is mentioned six times in this manuscript, including twice in the methods (i.e., woody category” and “perennial grass cover category).  I recommend checking this terminology for consistency throughout the manuscript.

Line 411:  Change “, also in North Dakota [39]” to “in a second North Dakota study.”

Line 441:  Change “sparrows” to “grasshopper sparrows.”

Line 459:  Change “and [52]” to “and Shaffer et al. [52]”

Lines 464-467:  It would be useful to include some discussion on the new, emerging paradigm for the management of native grasslands that considers heterogeneity as part of a healthy, functioning ecosystem.  There are many recent papers on this topic:

Fuhlendorf, S.D., and Engle, D.M., 2001, Restoring heterogeneity on rangelands—Ecosystem management based on evolutionary grazing patterns: BioScience, v. 51, no. 8, p. 625–632. https://doi.org/10.1641/0006-3568(2001)051[0625:RHOREM]2.0.CO;2

Fuhlendorf, S.D., and Engle, D.M., 2004, Application of the fire–grazing interaction to restore a shifting mosaic on tallgrass prairie: Journal of Applied Ecology, v. 41, no. 4, p. 604–614. https://dx.doi.org/10.1111/j.0021-8901.2004.00937.x

Fuhlendorf, S.D., Fynn, F.W.S., McGranahan, D.A., and Twidwell, D., 2017, Heterogeneity as the basis for rangeland management, in Briske, D.D., ed., Rangeland systems—Processes, management and challenges: New York, N.Y., Springer Nature, p. 169–196. https://doi.org/10.1007/978-3-319-46709-2_5

Fuhlendorf, S.D., Harell, W.C., Engle, D.M., Hamilton, R.G., Davis, C.A., and Leslie, D.M., Jr., 2006, Should heterogeneity be the basis for conservation? Grassland bird response to fire and grazing: Ecological Applications, v. 16, no. 5, p. 1706–1716. https://dx.doi.org/10.1890/1051-0761(2006)016%5B1706:SHBTBF%5D2.0.CO;2

Line 492:  Change “landscape heterogeneity” to “habitat heterogeneity.”

References

General:  There were several inconsistencies in the format of the citations within the References section.  I tried to point out the major ones, but there were some minor inconsistencies that I did not include.

Citation 2: Capitalize “Breeding Bird Survey.”

Citations 4 and 5:  These citations include hanging indents, but none of the other citations in the References section have hanging indents.

Citation 8: Do not capitalize the secondary words in the title of the article.

Citation 12: Capitalize the “S” in the journal name: “BioScience.”

Citation 19: Do not capitalize the secondary words in the title of the article.

Citation 21: Capitalize “Sandhills” for consistency.

Citation 24: Do not capitalize the secondary words in the title of this thesis.  Elsewhere in the References section, thesis titles are not capitalized.

Citation 27: Write out (define) “UNL.”

Citation 30: Remove hyphen in journal title “Midland.”  It should be one word without a hyphen.

Citation 37: Do not capitalize the secondary words in the title of the article (except for Sandhills and Nebraska).

Citation 38: Should the period after “Thesis” be a comma?

Citation 41: Capitalize “Neotropical.”

Citation 43:  This citation is an annual report, and the following published article would be a better and more accessible reference for this research:  Salo, E.D., Higgins, K.F., Patton, B.D., Bakker, K.K., Barker, W.T., Kreft, B., Nyren, P.E., Grazing intensity effects on vegetation, livestock, and non-game birds in North Dakota mixed-grass prairie. Proceedings of the North American Prairie Conference 2004, 19, 205-215.  https://digitalcommons.unl.edu/napcproceedings/88/

Citation 49: Capitalize “Neotropical.”

Citation 50:  This publication has been updated.  Please use the following citation:  Shaffer, J.A., Igl, L.D., Johnson, D.H., Dinkins, M.F., Goldade, C.M., Parkin, B.D., Euliss, B.R., The effects of management practices on grassland birds—Upland Sandpiper (Bartramia longicauda), chap. F of Johnson, D.H., Igl, L.D., Shaffer, J.A., and DeLong, J.P., eds., The effects of management practices on grassland birds: U.S. Geological Survey Professional Paper 1842, 2019, 20 p., https://doi.org/10.3133/pp1842F.

Citation 51:  This publication has been updated.  Please use the following citation:  Martin, J.W., Parrish J.R. Lark Sparrow (Chondestes grammacus), version 1.0. In Birds of the World (A. F. Poole and F. B. Gill, Editors). Cornell Lab of Ornithology, Ithaca, NY, USA. 2020. https://doi.org/10.2173/bow.larspa.01

Citation 52:  This publication has been updated.  Please use the following citation:  Shaffer, J.A., Igl, L.D., Johnson, D.H., Sondreal, M.L., Goldade, C.M., Parkin, B.D., Euliss, B.R., The effects of management practices on grassland birds—Lark Sparrow (Chondestes grammacus), chap. DD of Johnson, D.H., Igl, L.D., Shaffer, J.A., and DeLong, J.P., eds., The effects of management practices on grassland birds: U.S. Geological Survey Professional Paper 1842, 2021, 16 p., https://doi.org/10.3133/pp1842DD.

Funding

Line 501:  Write out “USDA.”  Only mentioned once in manuscript.

Data Availability Statement

Line 505:  Write out “UNL.”  Only mentioned twice in manuscript.

Tables

Table 1:  Sample sizes are provided on Lines 137-140.  It would be useful to include a footnote in this table indicating sample sizes.

Table 2:  Change the common name of “Greater prairie chicken” to “Greater prairie-chicken.”  That is, add a hyphen.  Also, the scientific name of the northern harrier has recently changed.  Change “Circus cyaneus” to “Circus hudsonius.”

Table 3:  For the species common names, use lower case for the second (or third) words in the name (e.g., Brown-headed cowbird).

Table 4: For the species common names, use lower case for the second (or third) words in the name (e.g., Brown-headed cowbird).

Table 4:  In the text, the β for grasshopper sparrow is listed as -0.745 (Line 351), but in the table it is listed as -0.74.  Should the value in the table be -0.75?

Author Response

Replies to Reviewer #2

Lines 16-17:  Please provide the three grazing systems that were evaluated:  “We estimated species richness and density of grassland birds for three grazing systems used on private ranches: long, medium, and short duration rotational grazing systems.”

--Updated as suggested.

Lines 20-21:  Please provide scientific names for grasshopper sparrow, western meadowlark, and brown-headed cowbird.

--Updated as suggested.

Line 21:  The authors provided a value for the percentage of all observations that comprised the three most common species, but there is a discrepancy on the percentage (70%) given on Line 21 in the Abstract and the percentage (approximately 75%) given on Line 316 in the Results.  Based on the additional information provided in the Results, the percentage is closer to 72.3%.  I recommend including this value in both places.

--Thank you. We updated with 72% in both places.

Line 22:  The wording (6 species’ density) at the end of this sentence is somewhat awkward.  I recommend rewording this sentence:  “We used a model comparison approach to determine the effects of habitat on the densities of six species.”

--Updated as suggested.

Introduction

Lines 45-46.  The sentence ends somewhat awkwardly. I believe that the second to last word of the sentence should be singular (i.e., region rather than regions):  “The erodible unformed soils have thwarted most efforts to cultivate crops and has left much of the region unconverted.”

--Updated as suggested.

Line 57 and 63:  Do not include a hyphen between “privately” and “owned.”  Compounds formed by an adverb ending in “ly” plus an adjective or participle (such as chiefly unrelated or neatly organized) are not hyphenated either before or after a noun.

--Updated as suggested.

Methods

Line 69: Do not include a hyphen between “privately” and “owned.”  Compounds formed by an adverb ending in “ly” plus an adjective or participle (such as chiefly unrelated or neatly organized) are not hyphenated either before or after a noun.

--Updated as suggested.

Line 128:  For consistency, write “3” as “three.”  For example, see Lines 137 and 139.

--Updated as suggested.

Line 139:  For consistency, write “twelve” as “12.” Numbers over 10 are likely written as numerals.

--Updated as suggested.

Line 146: For consistency, include a comma for “1,500-m” as done on Line 143.

--Updated as suggested.

Line 162:  Change the following “four-to-five hour period” to “4–5-hour period” because hour is a unit.

--Updated as suggested with hyphen between hour and period. We kept the ‘to’ as ‘4- to 5-hour’ which we believe is what is needed (4-hour and 5-hour combined). 

Line 179:  Provide the scientific name for “yucca.”

--Updated as suggested.

Line 181:  Change “Litter” to “The depth of litter.”

--Updated as suggested.

Line 199:  Write out “dm” to “decimeter.”

--Updated as suggested.

Line 210:  Delete “coefficient of variation” and just use “CV.”  Coefficient of variation was already defined on Line 45.

--Updated as suggested.

Lines 229-230:  Change “three-year means” to “3-year means” because year is a unit.

--Updated as suggested.

Line 255: Change “5” to “five” for consistency.

--Updated as suggested.

Results

Line 295:  Change “three years” to “3 years” because year is a unit.

--Updated as suggested.

Lines 296-297:  The sentence on these lines indicate that “Species richness estimates decreased with grazing duration (Fig. 3).”  In actuality (and as shown in the sentence on Lines 297-299 and Figure 3), species richness increased with grazing duration.  That is, for example, species richness was lower in short duration grazing systems (36.3 species) compared to long duration grazing systems (52.7 species).

--Sorry for our confusion. We were attempting to interpret the order of our figure (left to right). We have clarified in the text to address your point.

Line 316:  See comment above concerning “75%.”

--Updated as suggested.

Line 319:  Change “Dupland sandpiper” to “upland sandpiper.”

--Updated as suggested.

Line 350:  Insert a period after “density.”

--Updated as suggested.

Lines 364:  Should there be the word “and” between “Tables 3,4”?  That is, should it be written as “Tables 3 and 4?”

--Updated as suggested.

Line 373:  Delete “with.”

--Updated as suggested.

Discussion

Line 400:  Clarify.  The authors wrote: “Species richness tended to be higher on long duration study sites,…” Please clarify higher than what or compared to what.

--We have significantly rewritten this section to address multiple reviewer comments. We believe your concern is now addressed.

Line 406:  It is not clear what the authors mean by “meaningful ways among the same category.”  What is a meaningful way? What category are the authors referencing?  The same goes for Line 407.  What is a grazing system category? “Category” or “categories” is mentioned six times in this manuscript, including twice in the methods (i.e., woody category” and “perennial grass cover category).  I recommend checking this terminology for consistency throughout the manuscript.

--Wording was changed to better illustrate that our research was conducted on privately managed ranches and each ranch (and grazing system), was different in some way even if considered long, medium or short duration grazing.

Line 411:  Change “, also in North Dakota [39]” to “in a second North Dakota study.”

--Updated as suggested.

Line 441:  Change “sparrows” to “grasshopper sparrows.”

--Updated as suggested.

Line 459:  Change “and [52]” to “and Shaffer et al. [52]”

--Our understanding is that the journal does not use this format. We modified the sentence so we did not have to refer to the study as the noun of the sentence.

Lines 464-467:  It would be useful to include some discussion on the new, emerging paradigm for the management of native grasslands that considers heterogeneity as part of a healthy, functioning ecosystem.  There are many recent papers on this topic:

Fuhlendorf, S.D., and Engle, D.M., 2001, Restoring heterogeneity on rangelands—Ecosystem management based on evolutionary grazing patterns: BioScience, v. 51, no. 8, p. 625–632. https://doi.org/10.1641/0006-3568(2001)051[0625:RHOREM]2.0.CO;2

Fuhlendorf, S.D., and Engle, D.M., 2004, Application of the fire–grazing interaction to restore a shifting mosaic on tallgrass prairie: Journal of Applied Ecology, v. 41, no. 4, p. 604–614. https://dx.doi.org/10.1111/j.0021-8901.2004.00937.x

Fuhlendorf, S.D., Fynn, F.W.S., McGranahan, D.A., and Twidwell, D., 2017, Heterogeneity as the basis for rangeland management, in Briske, D.D., ed., Rangeland systems—Processes, management and challenges: New York, N.Y., Springer Nature, p. 169–196. https://doi.org/10.1007/978-3-319-46709-2_5

Fuhlendorf, S.D., Harell, W.C., Engle, D.M., Hamilton, R.G., Davis, C.A., and Leslie, D.M., Jr., 2006, Should heterogeneity be the basis for conservation? Grassland bird response to fire and grazing: Ecological Applications, v. 16, no. 5, p. 1706–1716. https://dx.doi.org/10.1890/1051-0761(2006)016%5B1706:SHBTBF%5D2.0.CO;2

            --Thank you for the suggestion. We followed your advice.

Line 492:  Change “landscape heterogeneity” to “habitat heterogeneity.”

Updated as suggested.

References: all updated as suggested

General:  There were several inconsistencies in the format of the citations within the References section.  I tried to point out the major ones, but there were some minor inconsistencies that I did not include.

Citation 2: Capitalize “Breeding Bird Survey.”

Citations 4 and 5:  These citations include hanging indents, but none of the other citations in the References section have hanging indents.

Citation 8: Do not capitalize the secondary words in the title of the article.

Citation 12: Capitalize the “S” in the journal name: “BioScience.”

Citation 19: Do not capitalize the secondary words in the title of the article.

Citation 21: Capitalize “Sandhills” for consistency.

Citation 24: Do not capitalize the secondary words in the title of this thesis.  Elsewhere in the References section, thesis titles are not capitalized.

Citation 27: Write out (define) “UNL.”

Citation 30: Remove hyphen in journal title “Midland.”  It should be one word without a hyphen.

Citation 37: Do not capitalize the secondary words in the title of the article (except for Sandhills and Nebraska).

Citation 38: Should the period after “Thesis” be a comma?

Citation 41: Capitalize “Neotropical.”

Citation 43:  This citation is an annual report, and the following published article would be a better and more accessible reference for this research:  Salo, E.D., Higgins, K.F., Patton, B.D., Bakker, K.K., Barker, W.T., Kreft, B., Nyren, P.E., Grazing intensity effects on vegetation, livestock, and non-game birds in North Dakota mixed-grass prairie. Proceedings of the North American Prairie Conference 2004, 19, 205-215.  https://digitalcommons.unl.edu/napcproceedings/88/

Citation 49: Capitalize “Neotropical.”

Citation 50:  This publication has been updated.  Please use the following citation:  Shaffer, J.A., Igl, L.D., Johnson, D.H., Dinkins, M.F., Goldade, C.M., Parkin, B.D., Euliss, B.R., The effects of management practices on grassland birds—Upland Sandpiper (Bartramia longicauda), chap. F of Johnson, D.H., Igl, L.D., Shaffer, J.A., and DeLong, J.P., eds., The effects of management practices on grassland birds: U.S. Geological Survey Professional Paper 1842, 2019, 20 p., https://doi.org/10.3133/pp1842F.

Citation 51:  This publication has been updated.  Please use the following citation:  Martin, J.W., Parrish J.R. Lark Sparrow (Chondestes grammacus), version 1.0. In Birds of the World (A. F. Poole and F. B. Gill, Editors). Cornell Lab of Ornithology, Ithaca, NY, USA. 2020. https://doi.org/10.2173/bow.larspa.01

Citation 52:  This publication has been updated.  Please use the following citation:  Shaffer, J.A., Igl, L.D., Johnson, D.H., Sondreal, M.L., Goldade, C.M., Parkin, B.D., Euliss, B.R., The effects of management practices on grassland birds—Lark Sparrow (Chondestes grammacus), chap. DD of Johnson, D.H., Igl, L.D., Shaffer, J.A., and DeLong, J.P., eds., The effects of management practices on grassland birds: U.S. Geological Survey Professional Paper 1842, 2021, 16 p., https://doi.org/10.3133/pp1842DD.

Reviewer 3 Report

Comments and Suggestions for Authors

See file below for minor suggested edits.

Author Response

Replies to Reviewer #3

A comment on semantics: The authors might want to consider the consistency of terms describing the long-, medium-, and short-duration periods. Some rangeland managers might consider these three periods as treatments that are all part of a rotational-grazing system, whereas some managers might consider the long-duration period as a traditional (season-long) grazing system and the short- and medium-duration periods as parts of a rotational grazing system. The authors refer to each period as systems (Example from the abstract: “We estimated species richness and density of grassland birds for three grazing systems used on private ranches”). The authors are pretty clear that they are considering each period as a system, until Line 419 in Discussion, when there is reference to rotational systems and cites the Kempema thesis. “Rotational systems were relatively more homogeneous as compared to long duration systems at both a small- and large spatial scale [24].” In that sentence, a distinction appears to be made between the short- and medium-duration periods as being parts of a rotational grazing system, and the long-duration period as being part of a different, but yet unnamed, grazing system. The authors should make this clear: are they considering all three periods as part of a rotational grazing system, or is there a traditional grazing system also involved? And, if this is the case, then do not refer to each period as a system. Again, because the authors have supplied all the details necessary for a range manager to implement the periods, or treatments, or systems, or whatever you may want to call them, this is just a matter of semantics and consistency. It doesn’t detract from a manager having the information he/she needs to understand the period/treatment/system.

--We have updated our wording describing these systems.  The ‘long duration’ is the ‘continuous’ system. It has the longest duration of grazing but is not rotational. Thank you.

Error in line 319. Remove the “D” before upland sandpiper

--Updated as requested

Line 350: Period missing between sentences in Line 350.

--Updated as requested

Lines 351 to 353: Findings for lower-ranked models pertaining to GRSP are awkwardly stated. Findings of lower-ranked models are better stated in paragraph below describing results for LASP (For example. “Lower-ranked models provided some evidence that lark sparrow densities decreased with more litter cover” starting in Line 357)

--Clarified as requested

Line 354 “on the study ranches” unnecessary. Seems out of place to call the study areas “ranches”

--Updated as requested

Line 361 to 363: odd to say that medium duration grazing systems “supported” the lowest densities. Might make more sense to say that short duration grazing systems supported the highest densities. Otherwise, a reader may think that low densities are the desired outcome.

--Clarified as requested

Table 4, rather than cut off a species’ name mid-way, move second part of name to next row (e.g., instead of splitting cow-bird, move the entire name to the next row). See also GRSP.

--Table format revised to address issue.

Line 406: Intriguing statement: “Grazing systems in our study varied in meaningful ways among the same category [21,37].” Not quite sure what this means to convey.

--This section has been revised extensively to address multiple reviewer comments. We believe it should be satisfactory now.

 Lines 406 to 411: These lines state: It is no surprise, therefore, that previous research shows a low level of agreement on grazing system effects. For example, grasshopper sparrow densities were similar among grazing system treatments in a North Dakota study [38], higher on rotational systems, also in North Dakota [39], and highest on long duration (season-long continuous) systems in Nebraska [40].”

The studies cited are three theses. The authors could also refer to Salo et al. 2004. Salo, E.D., Higgins, K.F., Patton, B.D., Bakker, K.K., and Barker, W.T., 2004, Grazing intensity effects on vegetation, livestock and non-game birds in North Dakota mixed-grass prairie, in Egan, D., and Harrington, J.A., eds., Proceedings of the nineteenth North American Prairie Conference: Madison, Wis., University of Wisconsin, p. 205–215.

As summarized in Shaffer et al. 2023 (version 1.1): In the northern mixed-grass prairies of North Dakota, Grasshopper Sparrow densities were higher in lightly to moderately grazed pastures than heavily or extremely heavily grazed pastures, and grasslands grazed at low-to-moderate rates had greater biomass reserves that benefitted the suite of grassland bird species while maintaining acceptable daily rates of gain for individual cattle (Salo and others, 2004).

--Thank you for that summary. We added the Salo citation elsewhere. In this sentence, we were explicitly looking for citations having to do with grazing systems, while Salo focuses on grazing intensity which is also an important point.

Lines 433 and 434. The authors provide supporting evidence for the relationships between shrub cover and densities of UPSA and LASP, but do not do so for GRSP. Although the finding of an increase in GRSP density with increasing shrub cover may seem counterintuitive, a number of other researchers have found a similar relationship. The authors could refer to Shaffer et al. 2023 (ver 1.1; reference below), which summarizes findings of similar relationships.

For example: “In tallgrass prairies of Nebraska and Iowa, Grasshopper Sparrow density and occurrence were positively related to shrub density (McLaughlin and others, 2014).

In a second study of reclaimed coal mines in Indiana, Grasshopper Sparrows occurred in habitats of open grasslands and shrub-savanna areas (defined as grasslands with many scattered young trees and shrubs) (Galligan and others, 2006).

Grasshopper Sparrows are generally considered woodland-intolerant species (Grant and others, 2004), but the species occupies grasslands in oak savanna and oak barren habitats (Rao and others, 2008). Grasshopper Sparrows also tolerate a moderate degree of short-statured shrubs within native prairies (Arnold and Higgins, 1986; Schneider, 1998; Henderson and Davis, 2014).

…in southeastern Arizona, Ruth and others (2020) found that shrubs were used by Grasshopper Sparrows as air temperature in desert grasslands increased. Ruth and others (2020) documented the use of shrubs as thermal refugia during periods of extremely high temperatures, such as when the difference between ground temperature in direct sun and ground temperature beneath shrubs differed by 15 degrees Celsius (ºC).

In southwestern Saskatchewan, Grasshopper Sparrow abundance increased with increasing shrub cover (Henderson and Davis, 2014).

Grasshopper Sparrows were present in grasslands with low coverage of shrubs >1 m tall. Indicated pairs declined as coverage of low shrubs increased (Grant and others, 2010). Grasshopper Sparrows were present more often in grasslands with low levels of quaking aspen (Populus tremuloides) woodland within 100 m of vegetation points than in unoccupied areas (Grant and others, 2004). In mixed-grass prairies in south-central North Dakota, Grasshopper Sparrows occurred along both shrubby and shrubless transects but were most abundant on the shrubless transects (Arnold and Higgins, 1986). Within grazed mixed-grass prairies in North Dakota, abundance of Grasshopper Sparrows was positively associated with density of low-growing shrubs (western snowberry [Symphoricarpos occidentalis] and silverberry [Elaeagnus commutata]) (Schneider, 1998),

In remnant tallgrass prairies in Minnesota, Elliott and Johnson (2017) reported a curvilinear relationship between Grasshopper Sparrow density and coverage of shrubs; Grasshopper Sparrow density peaked at about 4 percent shrub coverage.

Grasshopper Sparrow territories may include elevated structures that are used as song perches. In desert grasslands, Ruth and Skagen (2017) reported that male Grasshopper Sparrows regularly used taller shrubs within their territories as favored song perches, a pattern that was more obvious in grasslands with low shrub densities. Ruth and Skagen (2017) suggested that there is a low threshold of shrub density acceptable to Grasshopper Sparrows, especially males, and that selecting territories and nest sites in relation to shrub density may be a balancing act.

(All summarized from Shaffer et al. 2023 [version 1.1])

Shaffer, J.A., Igl, L.D., Johnson, D.H., Sondreal, M.L., Goldade, C.M., Nenneman, M.P., Wooten, T.L., and Euliss, B.R., 2021, The effects of management practices on grassland birds—Grasshopper Sparrow (Ammodramus savannarum) (ver. 1.1, May 2023), chap. GG of Johnson, D.H., Igl, L.D., Shaffer, J.A., and DeLong, J.P., eds., The effects of management practices on grassland birds: U.S. Geological Survey Professional Paper 1842, 59 p., https://doi.org/10.3133/pp1842GG.

--Thank you for these notes. We have added this reference to the suggested location.

Lines 458 to 463. Note that, in reference to these statements in quotations below, citation 52 has been updated (see comments below for References). “Bare ground is important for foraging habitat, and [52] recommended moderate grazing for lark sparrows. Although lark sparrows tend to select grazed areas [55], our data suggest high levels of grazing could reduce lark sparrow densities. Management for this species should include moderate levels of litter accumulation and bare ground [55]”

For example, the Capsule statement from LASP in Shaffer et al, 2021 now reads: “Keys to Lark Sparrow (Chondestes grammacus) management include providing open grasslands with sparseto-moderate herbaceous and litter cover and a woody component and allowing occasional burning or moderate grazing. Lark Sparrows have been reported to use habitats with 10–63 centimeters (cm) average vegetation height, 10–54 percent grass cover, 9–25 percent forb cover, 4– 18 percent shrub cover, 16–38 percent bare ground, 12–45 percent litter cover, and less than or equal to (≤) 1 cm litter depth.”

--Thank you for this update. This seems to match our statement.

Starting at Line 495: Author contributions: a semicolon is missing between “methodology, L.P., S.K. W.S. and formal analysis” A semicolon is also missing before “supervision”

--Updated as requested

References

Reference 50. The UPSA species accounts has been updated from this:

Dechant, J. A., Dinkins, M. F., Johnson, D. H., Igl, L. D., Goldade, C. M., Parkin, B. D., Euliss, B. R. Effects of management practices on grassland birds: Upland Sandpiper. Northern Prairie Wildlife Research Center, Jamestown, ND, 1999 (revised 2001); 33 pp.

To this: Shaffer, J.A., Igl, L.D., Johnson, D.H., Dinkins, M.F., Goldade, C.M., Parkin, B.D., and Euliss, B.R., 2019, The effects of management practices on grassland birds—Upland Sandpiper (Bartramia longicauda), chap. F of Johnson, D.H., Igl, L.D., Shaffer, J.A., and DeLong, J.P., eds., The effects of management practices on grassland birds: U.S. Geological Survey Professional Paper 1842, 20 p., https://doi.org/10.3133/pp1842F.

--Updated as requested

Reference 52. The LASP species account has been updated from this: 52 Dechant, J. A., Sondreal, M. L., Johnson, D. H., Igl, L. D., Goldade, C. M., Parkin, B. D., Euliss, B. R. Effects of management practices on grassland birds: Lark Sparrow. Northern Prairie Wildlife Research Center, Jamestown, ND, USA, 1999 (revised 2002); 18 pp. To This: Shaffer, J.A., Igl, L.D., Johnson, D.H., Sondreal, M.L., Goldade, C.M., Parkin, B.D., and Euliss, B.R., 2021, The effects of management practices on grassland birds—Lark Sparrow (Chondestes grammacus), chap. DD of Johnson, D.H., Igl, L.D., Shaffer, J.A., and DeLong, J.P., eds., The effects of management practices on grassland birds: U.S. Geological Survey Professional Paper 1842, 21 p., https://doi.org/10.3133/pp1842DD.

--Updated as requested

Reference 53. NOTE: The WEME account that is part of this series has yet to be updated, so the older reference is still ok.

--Thank you.